# Persistent Large Granular Lymphocyte Clonal Expansions: “The Root of Many Evils”—And of Some Goodness

**DOI:** 10.3390/cancers14051340

**Published:** 2022-03-05

**Authors:** Carlos Bravo-Pérez, Salvador Carrillo-Tornel, Esmeralda García-Torralba, Andrés Jerez

**Affiliations:** 1Hematology and Medical Oncology Department, University Hospital Morales Meseguer, IMIB, 30007 Murcia, Spain; salvacarri5@gmail.com (S.C.-T.); esme_bk@hotmail.com (E.G.-T.); anjecayu@gmail.com (A.J.); 2Centro de Investigación Biomédica en Red de Enfermedades Raras CIBERER-CB15/00055, 30003 Murcia, Spain

**Keywords:** large granular lymphocyte leukemia, autoimmunity, hematological disorders, solid tumors, transplantation

## Abstract

**Simple Summary:**

Large granular lymphocyte leukemia (LGLL) is a chronic disorder of either mature T or NK lymphocytes. As clonal expansions of the immune system cells, difficulties in the distinction between a true neoplasia and a physiological reactive process have been common since its description. We review here the different conditions associated with persistent clonal LGL expansions and discuss their potential origin and whether they can modulate the clinical features.

**Abstract:**

Large granular lymphocyte leukemia (LGLL) is a chronic disease of either mature phenotype cytotoxic CD3+ T lymphocytes or CD3- NK cells. LGLL diagnosis is hampered by the fact that reactive persistent clonal LGL expansions may fulfill the current criteria for LGLL diagnoses. In addition to the presence of characteristic clinical and hematological signs such as anemia or neutropenia, LGLL/LGL clonal expansions have been associated with an array of conditions/disorders. We review here the presence of these persistent clonal expansions in autoimmune, hematological disorders and solid neoplasms and after hematopoietic stem cell transplantation. These associations are a unique translational research framework to discern whether these persistently expanded LGL clones are causes or consequences of the concomitant clinical settings and, more importantly, when they should be targeted.

## 1. Introduction

Large granular lymphocyte leukemia (LGLL) is defined as an abnormal clonal expansion of mature LGLs that remain long-term competent [1]. However, a “clonal expansion of mature and long-term competent LGL” may also be a description of a physiological reactive process. Thus, the distinction relies on what we consider “abnormal”, but proposed criteria until now (persistence, absolute lymphocytosis, TCR rearrangement, immunophenotypic profile) have been seen in expansions considered reactive. The presence of an acquired mutation in the JAK/STAT pathway seemed to definitely separate neoplastic from reactive cases. However, as elegantly discussed elsewhere, the presence of cancer-related gene mutations in healthy organs has blurred the border between neoplastic and normal tissues [2]. Actually, somatic *STAT3* mutations have been recently detected in CD8 + T cells of healthy blood donors carrying human T-cell leukemia virus type 2 [3].

The diagnosis of LGLL, according to the 2016 classification of the World Health Organization (WHO), is based on a persistent (>6 months) increase in the number of LGL cells in peripheral blood (PB), usually 2–20 × 10^9^/L, without an identifiable cause. The WHO update has not introduced relevant changes in the diagnosis of this entity, and the division is maintained, according to the lineage of origin of the leukemic cell, in LGLL of T lymphocytes (T-LGLL, 85% of cases) and chronic lymphoproliferative disease of NK cells (CLPD-NK). The molecular hallmark of LGLL is the mutation of *STAT3* and other genes associated with the JAK/STAT pathway. Treatment is based on immunosuppressive therapy (methotrexate, cyclophosphamide, etc.), and it might enable disease control and long-term survival, but not leukemic clone eradication. Beyond the presence of characteristic clinical and hematological signs, such as neutropenia or anemia, LGLL has been associated with an array of conditions/disorders [4].

Since Loughran’s seminal report of LGLL in February 1985 [5], it took only 8 months for the first association to be reported with polyarthritis [6]. Loughran also first comprehensively compiled the different conditions associated with LGLL, and reviewed them under the soubriquet of “the root of many evils” [7]. We revisit here those evils, more described since, and a few associations related to good outcomes. We have kept the definition used by each author, even though many of the “LGL clonal expansions” may fulfill current WHO-defined LGLL criteria.

## 2. Autoimmune Manifestations

The association of LGLL with autoimmune disorders is well established [1,7]. However, the exact pathogenic connection between LGLL and autoimmunity remains unknown and, probably, is not a one-way process. Whilst in some cases the autoimmune manifestations may be secondary to pro-inflammatory cytokine release and cytotoxic activity from the leukemic clone [8,9], in others, autoimmunity may be the primary pathogenic phenomenon, so that the autoantigen-driven chronic stimulation of the immune system may also eventually lead to the lymphoproliferative disorder.

The landscape of autoimmune manifestations related to LGLL is wide and diverse (Table 1 and Table 2). A high proportion of patients with LGLL have laboratory findings supporting an underlying autoimmune context, mainly positivity for rheumatoid factor (40–60%) and antinuclear antibodies (30–40%), as well as autoantibodies directed against granulocytes (40–60%), red blood cells (15%) or platelets (10–30%). Moreover, in a significant proportion of these cases, serologic markers may be accompanied by overt autoimmune disease. Cytopenia, mainly immune-mediated, is the most frequent form of presentation of LGLL [7,10], and rheumatoid arthritis (RA) the most characteristic autoimmune entity associated with this disease [6,11,12,13,14].

### 2.1. Cytopenia(s), Underlying Immune-Mediated Mechanisms, and Prevalence of Peripheral Autoimmune Cytopenia(s)

Cytopenias, mainly neutropenia, but also anemia and thrombocytopenia, are the most common complications observed in patients with LGLL, and the main indications for treatment (Table 1) [12,13,14]. Not a single factor, but different pathogenic mechanisms of both central and peripheral origin may explain these hematological abnormalities in LGLL [10,15,16]. However, beyond lymphoproliferation and treatment-related toxicity, several pieces of evidence support that immune-mediated hematopoietic cell dysfunction and/or death are the preponderant processes underlying LGLL-associated cytopenia(s). On the one hand, BM infiltration by LGLL clones is characteristically mild and does not correlate with the grade of cytopenia(s) commonly appreciated in LGLL [16]. Indeed, a BM exam usually reveals slight hypercellularity with left-shifted maturation of granulocytic and erythroid precursors [17]. Similarly, despite the presence of splenomegaly in 25% of cases [12,14], it is frequently mild and also not proportional to the severity of cytopenia(s) [18]. On the other hand, the high serological positivity rate for autoantibodies directed against blood cells, together with the documented contribution of other immune mechanisms [19,20,21,22], suggest that enhanced hematopoietic cell destruction, by both humoral and cell-mediated immune responses, significantly underlies the development of LGLL-associated cytopenia(s). However, it should be also mentioned that it is still unclear whether the detection of autoantibodies or other immune markers may unfailingly cause the hematological abnormalities, or might also be a “paraphenomenon” of altered/cross-reactive immunity in some cases.

Acquired BM failure (BMF) syndromes and autoimmune cytopenias represent the paradigms of central and peripheral hematopoietic cell destruction, respectively, that can be observed in symptomatic LGLL patients. BMF syndromes are characterized by the immune attack against hematopoietic stem cells and variable-grade cytopenia; aplastic anemia (AA), pure red cell aplasia (PRCA), paroxysmal nocturnal hemoglobinuria (PNH), and low-risk myelodysplastic syndromes (MDS) are included in this group [23]. Autoimmune cytopenias are dominated by autoantibody-mediated destruction of peripheral blood cells; they include autoimmune neutropenia, autoimmune hemolytic anemia (AIHA), immune thrombocytopenic purpura (ITP), and multi-lineage immune cytopenias [24]. However, in most cases with LGLL-associated cytopenia(s), the border between these two “central” and “peripheral” mechanistic ends may be blurred. It is likely that differences in the cellular targets (stem vs. lineage-restricted cells) and/or in the modes of hematopoietic inhibition by the immune system may explain the *continuum* of LGLL-associated cytopenia(s), but these puzzling questions, as well as the exact pathogenic role of LGLL clones, remain unclear. Nevertheless, lessons from AA and from secondary autoimmune cytopenia(s) suggest that both the leukemic clone and the BM microenvironment in which it exists may be involved in the pathogenesis of LGLL-associated cytopenia(s) [25,26], and that immune-mediated hematopoietic failure observed in these cases may be due to the variable contribution of the following factors: (1) infiltration by LGLL clones and direct/induced cytotoxic T-cell-mediated destruction [27]; (2) increased secretion of proinflammatory cytokines, including IFN-α2, IFN-γ and IL-15, which may result in hematopoiesis inhibition [23]; and (3) autoantibody-dependent toxicity directed against blood cells [25].The association of LGLL with BMF syndromes will be further reviewed in Section 3: “Bone marrow failure and other hematologic neoplasms”.

Neutropenia is the most common presentation in LGLL (60–80%) [12,14], and recurrent/severe infections remain as one of the main causes of morbidity and death, as well as of treatment initiation [28,29]. Anti-neutrophil autoantibodies suggesting autoimmune neutropenia are evidenced in 40 to 60% of cases, and increased levels of granulocyte-apoptotic signals have also been reported [22]. Immunosuppressive therapy may result in absolute neutrophil count improvement, and supportive treatment with the use of colony-stimulating factors might also be explored as a complementary measure in some situations [13,29]. Anemia is the second abnormality most frequently observed, in up to 50% of cases [30]; its onset in LGLL cases may be attributed to a wider array of mechanisms [13,31]. Anti-erythrocyte autoantibodies are detected in nearly 15% of cases, and the estimated prevalence of AIHA ranges from 5 to 9% [30]. Thrombocytopenia mainly of mild-moderate grade is reported in 10 to 30% of cases, and anti-platelet autoantibodies, suggesting peripheral platelet destruction, are positive in a similar proportion of patients [30]. Moreover, cases of severe megakaryocytic thrombocytopenia resembling ITP have been reported in LGLL [12,32]. Additionally, and similarly to chronic lymphocytic leukemia, the synchronic or metachronic combination of AIHA and ITP, characteristic of Evans’ syndrome, has been reported in few cases [33,34].

### 2.2. Rheumatoid Arthritis and Felty’s Syndrome

RA is the most common inflammatory joint disease in the general population (0.5–2%) [35], and the autoimmune trait most frequently diagnosed in patients with LGLL (10–30%). It is a chronic systemic disease primarily involving the synovial tissue, dominated by joint inflammation and destruction. Extra-articular manifestations can occur in up to 40% of RA patients; they are more common in cases with elevated titers of rheumatoid factor, as well as in those with prolonged and aggressive clinical courses [36]. Hematological abnormalities, mainly anemia, but also other cytopenias, and less frequently lymphadenopathy and splenomegaly, are extra-articular manifestations of RA [37]. A small proportion of RA cases (<1%) develops Felty’s syndrome (FS), a severe complication described by Felty in 1924 that is defined by the triad of RA, neutropenia and splenomegaly [38]. Interestingly, the prevalence of FS in LGLL, although not consistently reported, is thought to be similar, or even higher, than that of FS in RA [11]. Furthermore, LGL expansions have been reported in 4% of cases with RA [39], and in 30 to 40% of patients with FS (Table 2) [40,41].

Most works associating LGLL with cytopenias, RA and relative disorders predominantly refer to the T-LGLL subtype. However, several series have performed subgroup analysis comparing clinical manifestations of LGLL according to the T or NK cell origin, and most of them state that, although less frequent in absolute counts, CLPD-NK is similarly linked to immune-mediated cytopenias and autoimmune manifestations.. Nevertheless, the low number of cases with CLPD-NK reported in each individual work precludes additional strong conclusions [12,60]. Of note, Poullot et al. in 2014 specifically conducted a multicenter, pooled analysis of 70 CLPD-NK cases. Despite the high similarity to T-LGLL in terms of demographics and immune manifestations, CLPD-NK was less likely to become symptomatic (18% vs. 49%), had a lower prevalence of severe neutropenia (33% vs. 61%) and was associated with RA more rarely (7% vs. 17%) [61].

The impact of *STAT3* mutational status on clinical immune manifestations has also been assessed in several studies [54,62]. In the pivotal work by Koskela et al. in 2012, which discovered *STAT3* mutations in LGLL, *STAT3*-mutated patients, compared to *STAT3*-wild-type ones, were more likely to have neutropenia (77% vs. 50%) and RA (26% vs. 6%) [54,62]; these findings were shortly validated by Jerez et al. [60]. More recently, Sanikommu et al., in an updated analysis of clinical features and outcomes in 204 LGLL patients, with 66 out of 183 (36%) carrying mutations in *STAT3*, communicated that *STAT3*-mutated patients were more likely to develop neutropenia (62% vs. 37%), anemia (49% vs. 34%) and RA (29% vs. 9%) than *STAT3*-wild-type subjects [14]. Interestingly, Rajala et al. analyzed the clonal diversity of *STAT3* mutations in LGLL in the same cohort; they found that 17% of the *STAT3*-mutated patients harbored multiple mutations, and that compared to *STAT3*-wild-type patients, RA was significantly more common in patients with T-LGLL carrying single (23% vs. 6%) or multiple *STAT3* (43% vs. 6%) mutations [63]. Altogether, these results point towards the involvement of chronic immune stimulation in the pathogenesis of LGLL.

Cumulative evidence demonstrates that the association of LGLL and RA/FS is unquestionable, but going further, it has been suggested that both processes may have a common etiopathogenesis. Basically, both entities have been shown to harbor oligoclonal or clonal expansions of effector T cells with memory phenotype [50,64], aberrant expression of NK cell markers (CD57+, CD94/NKG2+) [65,66], as well as constitutive activation of these cells via the JAK/STAT pathway, dominated by dysregulation of *STAT3*-targeted genes (*SOCS3, BCL3, PIM1*), overproduction of cytotoxic [19,67] and pro-inflammatory molecules [20,68], and defective response to pro-apoptotic signals [21,69,70,71]. Antigen-driven immune stimulation, induced or not by a primordial viral agent, and favored in a human leukocyte antigen DR4 (HLA-DR4) context, which is highly prevalent in both LGLL and RA [72,73], might initially underlie polyclonal/oligoclonal LGL expansion. Eventually, acquisition of somatic mutations, mainly in *STAT3* and other genes of the JAK/STAT pathway [54,62], may initiate and promote the monoclonal expansion observed in LGLL and explain, from this point forward, the divergent but parallel progression of LGLL and RA within a patient (Figure 1).

Furthermore, the overlap between LGLL with RA and FS is almost total [4,62,74], and so, it has been speculated that LGLL with RA and FS are elements of the same disease spectrum. This hypothesis gains even more strength when considering the following observations: (1) *STAT3* mutational status is closely associated with the presence of RA in patients with LGLL; (2) *STAT3* mutations, revealing “hidden” cases of LGLL, have been detected in 6 out of 14 (43%) subjects with prior diagnosis of FS, as reported by Savola et al. [40]; and (3) “aleukemic” forms of T-LGLL, with severe neutropenia and splenomegaly, some of them previously misdiagnosed with FS, have been recently reported by Gorodetskiy et al. LGL infiltration and clonal TCR rearrangement were detectable in splenic tissue in all patients, and *STAT3* was mutated in 3 (30%) of them. However, no LGLL clone was evidenced in PB by morphologic/immunophenotypic examination in any case, and only by molecular methods in 3 (30%) patients [75].

### 2.3. Other Autoimmune Diseases Associated with LGLL

Several case reports reveal many other autoimmune disorders, both systemic and organ-specific, anecdotally observed in patients with LGLL (Table 2). Sjögren’s syndrome has been variably associated; it can be a primary rheumatic or secondary, mainly to RA, disease that characteristically targets exocrine glands and provokes sicca syndrome [76]. The prevalence of Sjögren’s syndrome in LGLL is highly heterogeneous among series, from only 2% to up to 27% [12,13,42]. However, such differences could be explained by the fact that many cases of sicca manifestations might be subclinical [42]. Lupus erythematosus [43,44,45,46], systemic vasculitis [47], Behçet disease [48], inflammatory myositis [49], and other rheumatic forms of polyarthritis [12], have also been rarely reported in LGLL.

A wide range of organ-specific autoimmune disorders have occasionally been linked to LGLL. Endocrinopathies are the most frequently reported, particularly autoimmune (Hashimoto’s) thyroiditis and its subclinical variants. Differences in detecting those subclinical entities may explain why the prevalence of thyroid disease varies largely among different LGLL study cohorts [12]. Gastrointestinal autoimmune disorders have also been increasingly reported. Other organ-committed syndromes have rarely been associated with LGLL, including neurologic disorders (multiple sclerosis, polyneuritis) [47,52], vascular lung and renal disease (pulmonary hypertension, glomerulonephritis) [12,47,53], inflammatory skin manifestations (leukocytoclastic vasculitis, pyoderma gangrenosum) [12], and acquired coagulopathy (hemophilia and/or hypofibrinogenemia) (Table 2) [56,57,58,59].

## 3. Bone Marrow Failure and Other Hematologic Neoplasms

LGLL has been consistently associated with other hematological disorders such as BMF syndromes (AA, PRCA, PNH, MDS), with a frequency of 3–10%, and other hematological malignancies [77,78,79,80].

### 3.1. Myelodysplastic Syndromes

MDS are defined according to the revised WHO classification (2016) as a heterogeneous group of clonal disorders of hematopoietic stem cells (HSC), characterized by cytopenias and an increased rate of progression to acute myeloid leukemia (AML) [81]. The co-occurrence of MDS with LGLL has been reported in several of the cohorts by various groups in recent decades (Table 3). Other authors did not establish a WHO-defined diagnosis and addressed them as LGL expansions.

Saunthararajah et al., in a cohort of 100 patients with an initial diagnosis of either MDS or T-LGLL, found that 9% of patients had concomitant T-LGLL/MDS. They observed that the absolute count of cells with LGL morphology was higher when T-LGLL was the unique disorder than in the subgroup of patients with both disorders (*p* < 0.05). They also showed a lower rate of responses to immunosuppression in patients carrying both entities, compared to the group with T-LGL alone (*p* < 0.01). They contemplated that older age and a higher burden of lesions in hematopoietic stem cells in MDS patients could be responsible for poorer responses [79].

After the discovery of acquired *STAT3* and *STAT5b* variants in LGLL, several groups investigated the mutational profile in the context of a concomitant BM disease. Jerez et al. studied an extensive cohort with 367 patients with MDS and 140 with idiopathic AA. Of the 367 patients diagnosed with MDS, 24 (6.5%) had a concomitant LGLL, of which nine (37.5%) had acquired *STAT3* mutations. Interestingly, they found that 2.5% of the patients (9/343) with MDS without a suspicion of LGLL also had clones mutated for *STAT3*. In two patients with MDS and one patient with AA, they confirmed the *STAT3* variant to be harbored exclusively by the LGL compartment (CD3+ CD8+ CD57+). MDS patients with mutated *STAT3* were significantly characterized by a higher frequency of BM hypocellularity and neutropenia. No differences in overall survival (OS) were observed between *STAT3*-mutated and *STAT3*-wild-type MDS/AA-LGLL. Finally, a panel of 17 genes recurrently mutated in myeloid neoplasms were studied in the *STAT3*-mutated MDS-LGLL group, finding a relevant frequency of somatic *NRAS* variants in about 50% of the cases [82].

Seven years later, the Cleveland group used a complementary approach, checking whether their 240 LGLL patients harbored myeloid neoplasm-acquired mutations. Using next generation sequencing (NGS), they found that the frequency of mutations in *STAT3/STAT5b* was 39% in patients with LGLL alone and 15% in patients with a concomitant diagnosis of MDS-LGLL. Interestingly, they found somatic mutations in “myeloid genes” in 26% of LGLL patients with no evidence of MDS. Strikingly, both the number of mutations per patient (1.7) and the variant allele frequency (35 ± 1.6%) were higher than expected in a cohort not diagnosed with a myeloid disorder. Additionally, there was an overrepresentation of clonal hematopoiesis of indeterminate potential (CHIP)-related mutations. This suggests that some cases of T-LGLL may coexist with CHIP or clonal cytopenia of undetermined significance (CCUS) and even with a non-diagnosed MDS. When LGLL and CHIP/MDS coexist, the clonal burden of the myeloid disease appears to be lower than that of LGLL, with the authors hypothesizing about the LGL clone keeping the myeloid one at bay by means of a tumor surveillance role [83].

On the other hand, Ai et al. studied a cohort of 721 patients diagnosed with MDS, of which 10 had concomitant MDS-LGLL disease (seven patients with T-LGLL, two with mixed phenotype and one with CLPD-NK). In these 10 patients, they sequenced 114 genes using NGS, with *ASXL1* (30%) and *STAG2* (30%) as the most frequently mutated genes in this setting [85]. Acquired variants in genes such as *STAT3* (20%), *STAT5B* (10%), *TNFAIP3* (10%) and *PTPRT* (10%) were also found, as previously reported [86,87,88,89,90]. As *TNFAIP3* and *PTPRT* are genes included in the JAK/STAT pathway, they could be responsible for the constitutive activation of anti-apoptotic pathways in *STAT3/STA5b*-wild-type cases [85].

Fewer observations have been made regarding the association of the NK cell counterpart of LGLL with MDS. Jerez et al. studied the distribution of *STAT3* mutations in a cohort of 50 patients with CLPD-NK, finding two cases (4%) in which MDS diagnosis had been made [60]. Ai et al. reported an additional case of concurrent MDS-CLPD-NK that was wild-type for *STAT3* but had an acquired mutation in *PTPRT* [85].

Two recent works have explored the genomic link between CLPD-NK and myeloid malignancies. Olson et al. recently reported, using whole-genome sequencing, that the NK compartment in CLPD-NK, was enriched in *TNFAIP3* (10%) and *TET2* (28%) mutations. Thrombocytopenia and resistance to immunosuppressive agents were uniquely observed in those patients with only *TET2* mutation [91]. To explore the mutational clonal hierarchy, Pastoret et al. performed whole-exome sequencing of sorted, myeloid, T, and NK cells in 46 CLPD-NK cases and found that *TET2* mutations were shared by myeloid and NK cells in three of four cases. They hypothesized that *TET2* mutations emerge in early hematopoietic progenitors, suggesting a common origin for CLPD-NK and myeloid malignancies [92].

Regarding studies with MDS and LGL clonal expansions, in a letter commenting Pastoret’s work, Komrokji et al. reported the largest cohort of MDS (N = 1177) in which the presence of a coexisting LGL clone was studied. An LGL clonal expansion was defined by a flow cytometry profile compatible with the presence of an activated T-cell phenotype, with no *STAT3* status available. Coexistence of an MDS and an LGL clone was found in 322 patients (27%), where TCR (γ/β) rearrangement was detected in 92% versus 24% in the subgroup with MDS alone. After analyzing both subgroups, they concluded that there were no differences in terms of OS between MDS patients with or without an LGL clone, regardless of the risk of the disease according to the IPSS-R. They also found no differences in relation to progression to AML (19% in both subgroups). At the same time, no significant differences were reported in terms of response rates to treatment, either supportive or modifying. Interestingly, *U2AF1* mutations were more commonly observed among MDS patients with an LGL clone than in those without (*p* = 0.047) [84].

In a patient with unexplained cytopenias, the potential presence of an MDS and/or LGLL must be assessed. However, the appropriate final diagnosis may be a challenge, particularly in those cases lacking solid MDS features (i.e., typical cytogenetics, excess of blasts, ring sideroblasts) or, in the case of LGLL, lymphocytosis. In addition, both diseases exhibit clinical and pathologic overlap with reactive conditions. It has been shown how *STAT3*-mutated LGLL clones were missed in MDS patients and, for some of them, reviewing the initial workup found that none met the diagnostic criteria for an MDS [82,93]. On the other hand, as stated above, an LGLL diagnosis does not exclude the presence of a concomitant MDS. In this setting, NGS has a clear beneficial role in routine diagnostic practice. The presence of “myeloid” acquired mutations with a higher variant allelic frequency is supportive for the diagnosis of MDS, and the inclusion of *STAT3* and *STAT5b* on the NGS panel provides evaluation for cryptic or unsuspected LGL leukemia [93,94]. In addition, a new flow cytometry approach has been shown to be more specific for T malignancies than the TCR rearrangement. The pattern of expression of the constant region 1 of the T-cell receptor β chain (TRBC1) has proved to be an accurate and simple method for assessment of the neoplastic nature of Tαβ cells in patients with a suspected T-LGLL [95,96].

#### Aplastic Anemia

Until 2003, only 12 cases with a concomitant diagnosis of AA and LGLL had been reported [97]. It is a common notion that LGLL is an underdiagnosed entity. In addition, the presence of pancytopenia may add complexity to the diagnosis, posing a challenge for the identification of these two entities coexisting. The widespread use of flow cytometry and advances in molecular diagnostics have increased the number of AA-LGLL cases.

AA is defined as a BMF syndrome affecting all blood cell lineages, with a predominant autoimmunity pathogenesis, in which CD8+ T cells participate in the elimination of hematopoietic stem cell precursors [98,99]. There is evidence of an expansion of memory T lymphocytes with TCR Vβ oligoclonality and CDR3 homology, highlighting the pathogenic role of aberrantly activated T cells [100,101,102].

Go et al. reported that 9 out of 203 (4.4%) LGLL cases had a concomitant diagnosis of AA. These patients were similar to those with isolated acquired AA in terms of clinical characteristics, other than the increase in lymphocytes with LGL morphology in PB and an excess of CD3+ CD8+ CD57+ cells in BM. Regarding treatment, the response to immunosuppressive therapy in the AA-LGLL group was unsatisfactory (median survival of 40 months) and five patients died of the disease [77].

Regarding the mutational profile in patients with AA-LGLL, Jerez et al., in their cohort of 140 patients with AA, found 11 (7.9%) cases with known LGLL, of which six (54.5%) had *STAT3* mutations, along with 10 (7.1%) patients mutated for *STAT3* without suspicion of LGLL. *STAT3*-mutated patients were characterized by a higher frequency of moderate severity disease (29% vs. 21%) and by responding better to first-line therapy (81% vs. 21%).

Some studies assessed AA and LGL clonal expansions. Zhang et al. studied the T-cell compartment in a cohort of 41 patients with AA and 46 diagnosed with hypocellular MDS. They observed a significant reduction in the CD4+/CD8+ ratio in both subgroups, compared to controls. Furthermore, the hypocellular MDS group was characterized by an increase in NK cells (CD3− CD16+ CD56+), T-LGL cells (CD3+ CD57+), and percentage of immature cells, and a reduction in B lymphocytes, compared to the AA group [103].

Recently, Lundgren et al. studied the somatic mutations profile in the T-cell compartment of 24 patients with AA and compared it with 20 healthy controls. They applied a customized panel of 2533 genes, detecting an enrichment of acquired mutations in AA patients´ T cells compared with healthy donors, mainly within the JAK/STAT and MAPK pathways [104], both key immune regulatory pathways [105,106,107]. Mutations in *STAT3* (2/24, 8.3%) were limited to the T-cell compartment in AA patients, as previously reported [54,82]. Interestingly, they showed how the mutation burden correlated with the CD8+ T-cell clonality within AA patients but not in controls, and that clonal hematopoiesis mutation transmission to T cells was frequently observed [104].

The expansion of the NK cell compartment, in the context of AA, was recently studied by Li et al. in a cohort of 50 patients diagnosed with non-severe AA. They found that the percentage of CD56^bright^ NK cells was significantly higher in these patients compared to healthy controls (*p* = 0.011). They also observed an overexpression of *NKG2D* and a lower expression of CD158a on NK cells and higher levels of cytokines (IL-2 and IL-18) in the serum of patients with non-severe AA, as markers of an overactivated innate immune system [108].

### 3.2. Pure Red Cell Aplasia

PRCA is a syndrome characterized by anemia with severe reticulocytopenia, in which the production of erythroid precursors is almost or totally compromised, but with a preserved granulopoiesis and megakaryopoiesis [109,110,111]. Within the PRCA classification, that associated with lymphoproliferative disorders such as LGLL, is considered a “secondary acquired PRCA” [110].

In a cohort of 203 T-LGLL patients at the Mayo Clinic, concomitant PRCA disease was diagnosed in 15 (7.3%) subjects [8]. Most PRCA-LGLL cases reported to date involved the T-cell lineage subtype, but CLPD-NK has also occasionally been reported [112,113].

In their 2003 review, Go et al. already underlined the difficulty in diagnosing simultaneously both entities, as splenomegaly and lymphadenopathies were uncommon, and 22% of cases did not show absolute lymphocytosis. In their analysis of 65 PRCA-LGLL patients, more than half achieved durable responses to cyclophosphamide and/or cyclosporine [97]. Some reports have suggested cyclophosphamide as a preferred first line agent when PRCA/LGLL coexist. Tabata et al. reported a case with a concomitant PRCA-T-LGLL in which cyclophosphamide was an effective treatment as long as the LGL clone was present [114]. Handgretinger et al. described a patient with PRCA in which a clonal expansion of T-LGL was detected, who was refractory to cyclosporine but responded when switching to cyclophosphamide [115].

Ishida et al. reported an association between LGLL cases mutated in the SH2 domain of *STAT3* and the coexistence of PRCA. They found that *STAT3*-mutated LGLL cases (23/53, 43.4%) were associated with a higher frequency of concomitant PRCA disease (15/23, 65%) compared to *STAT3*-wild-type patients (7/30, 23.3%) [116]. In a recently published study, Kawakami et al. detected that 43% (18/42) of the patients in their cohort with PRCA had mutations in *STAT3*, which was associated with a lower age of onset and a higher rate of resistance to cyclosporine compared to the group without mutations in *STAT3* [117]. Finally, Qiu et al. observed that those PRCA-LGLL patients with *wild-type STAT3* responded better to monotherapy with methotrexate or in combination with prednisone [118].

Schützinger et al. reported the case of a Caucasian patient with PRCA-T-LGLL and a TCRγ/δ rearrangement, who showed resistance to first-line treatment with cyclosporine and methotrexate. They started treatment with low doses of alemtuzumab, achieving a remission of the PRCA without serious toxicological complications except for an opportunistic infection by herpes zoster [119]. Other reported studies incorporated alemtuzumab into the therapeutic algorithm of patients with LGLL and PRCA [120,121,122].

Lacy et al. identified that the subgroup of patients with concomitant PRCA-T-LGLL had better responses to immunosuppressive agents than PRCA cases without the association, with both cyclophosphamide and cyclosporine being effective [123]. Similar results were extracted from a national Japanese study analyzing the long-term responses of a cohort of 185 PRCA patients, where 14 cases (7.6%) with a concomitant LGLL were identified [124].

Clonal expansions of the NK population have been observed in the PB in patients with PRCA, suggesting a hyperreactivity of these cells in the context of the disease. In addition, the combination therapy of glucocorticoids with prednisone as maintenance therapy showed promising results in these patients [112,113].

### 3.3. Paroxysmal Nocturnal Hemoglobinuria

PNH is defined as a complement-mediated hemolytic anemia, in which a clonal expansion of hematopoietic stem cells with somatic mutations in the *PIGA* gene is observed [125,126]. This alteration results in blood cells deficient for glycosylphosphatidylinositol (GPI)-anchored-proteins, which together with the absence of CD55 and CD59, promote erythroid lysis [127]. PNH cases have been reported to be associated with AA and, to a lesser extent, with neoplastic entities such as LGLL [78,98].

In relation to the co-occurrence of both entities (PNH-LGLL), there are not many reported cases. In 2001, Karadimitris et al. published the case of a PNH patient in whom a clonal expansion of T-LGL cells was detected, as a result of a possible antigenic stimulation in the context of the underlying disease [128]. In their cohort of 24 PNH patients, Risitano et al. found that four presented a T-LGL clonal expansion of the cytotoxic compartment. These results suggest a possible immune escape mechanism that favors the expansion of the PNH clone [78].

Other cases have shown oligoclonal or non-clonal LGL expansions associated with PNH. In 2006, Fukumoto and Gotlib reported a PNH patient with oligoclonal T-LGL expansion, accompanied by a monoclonal IgG-lambda gammopathy of undetermined significance [129]. Boyer et al. communicated the case of a concomitant PNH-T-LGL disease, confirmed by immunophenotypic analysis (CD3+ CD8+ CD56+ CD16+) and BM biopsy (interstitial infiltration of CD8+ T lymphocytes, with overexpression of TIA-1). They did not observe TCR rearrangements or *STAT3* mutations in the T-LGL compartment [130].

Current PNH therapy is based on complement inhibition (C5), with eculizumab and ravulizumab approved by the Food and Drug Administration (FDA) and the European Medicines Agency (EMA) [131,132]. Given the low frequency of a PNH-LGLL diagnosis, reports addressing responses to complement inhibition are anecdotal. Boyer et al. observed persistent neutropenia after treatment with eculizumab in a PNH patient with expansion of a leukemic T-LGL clone [130].

Interestingly, the NK compartment in PNH has been described as hyporeactive and quantitatively reduced in [133,134]. Howe et al. showed a significant decrease in KIR receptors (KIR-2DS1 and KIR-2DS5) in patients with AA and PNH, postulating an immunogenic association between both pathologies [135]. These findings have been related to a greater susceptibility to infections in these patients, due to hypo-responsiveness of NK cells [136].

### 3.4. Other Hematological Neoplasms

In the context of myeloproliferative disorders, a case with essential thrombocythemia and subsequent expansion of a T-LGL clone was reported after treatment with hydroxyurea [137]. Malani et al. identified a patient with acute myeloid leukemia and a T-LGLL, treated with combined chemotherapy based on cytosine arabinoside and daunorubicin, with an AML partial response but persistence of the T-LGL clone [138]. In 2017, a patient with acute promyelocytic leukemia (APL) was reported, in whom T-LGLL was diagnosed while on APL maintenance therapy, causing neutropenia. The revision of the initial BM biopsy found that a 10% T-LGL infiltration was already present [139].

Regarding the association of chronic myeloid leukemia (CML) and LGLL, several studies have described a marked peripheral lymphocytosis after treatment with dasatinib, fulfilling the criteria for a T-LGLL clone [140,141,142,143,144,145]. Mustjoki et al. observed a clonal expansion of T/NK cells during therapy with this tyrosine kinase inhibitor and showed that the group of patients with lymphocytosis had a better prognosis [140]. In a later study from the same group, they analyzed a cohort of 34 CML patients, of whom 20 were treated with dasatinib and 14 with imatinib. They detected a clonal expansion of both the cytotoxic T compartment and NK cells in the dasatinib group, but not in the imatinib group. These results suggested that these clones with cytotoxic activity could favor the elimination of CML cells [141]. This phenomenon is not exclusive to dasatinib, as T expansions have been reported during treatment of Philadelphia chromosome-positive acute lymphoblastic leukemia with nilotinib [146]. A case of chronic myelomonocytic leukemia and T-LGLL has also been reported, defined by flow cytometry of abnormal populations of monocytes and CD8 + T cells [147].

Finally, concomitant diagnoses of LGLL disease with various B-cell neoplasms such as chronic lymphocytic leukemia [148,149], hairy cell leukemia [150] or mantle cell lymphoma [151,152,153], among others, have also been occasionally reported.

## 4. Solid Neoplasms

Several series and reviews have noted that the association of LGLL with neoplastic processes might not be restricted to hematological cancers, but include solid tumors as well [154]. However, evidence about the real association of both disorders is heterogeneous.

Some works initially described a proportion of non-hematological malignancies in LGLL below 10% [12,51]. Bareau et al. reported in 2010 the coexistence of solid tumors in 10/229 (4.4%) LGLL patients [12]. The authors noted the insufficient evidence for linking both disorders, since most patients were over 60 years of age and thus had a higher risk of developing another malignancy. Nevertheless, later works, also conducted in large cohorts, communicated significantly higher rates of solid neoplasms in LGLL. In 2015, Viny et al. reported that solid tumors were present in 27/156 (17.3%) LGLL patients [155], an elevated prevalence that was thereafter confirmed in an updated analysis of this cohort (30/204, 14.9%). Notably, *STAT3* mutations were identified in 12/30 (40%) of these cases, and the distribution of solid tumors between *STAT3*-mutated and wild-type LGLL was not different in this work [14]. Recently, in 2021, Dong et al. reported similar results, with a proportion of 66/319 (20.7%) solid cancers in their LGLL cohort [13]. Remarkably, they noted an elevated frequency of solid cancer in NK-CLPD (10/24, 42%), higher than that observed in T-LGLL subtype (56/295, 19.0%) [13].

Overall, the most common primary tumor localizations, reported in 77/103 cancer patients from the three aforementioned largest LGLL cohorts, were: prostate (*N* = 22), breast (*N* = 20), lung (*N* = 10), skin (*N* = 10, mainly melanoma), colorectal (*N* = 9), and kidney (*N* = 6). Of note, in most cases, cancer diagnosis was prior to LGLL identification [12,13,148].

There is not a clear explanation for the heterogenous rates reported, but differences in baseline patient characteristics, cancer types, antineoplastic treatments, and many other variables among works might be implied [12,13,148].

Despite unquestionable increasing evidence, there are not much data on the pathogenic association between LGLL and solid neoplasms, as well as on the therapeutic and prognostic implications of LGLL diagnosis in cancer patients. Some authors have hypothesized that the tumor may act as an antigenic trigger, leading to chronic immune stimulation. LGL expansion could additionally be favored in this context by immune system disturbances secondary to antineoplastic/immunosuppressive treatments. Moreover, LGLL diagnosis in patients with active cancer may entail a clinical challenge, particularly if severe or prolonged cytopenias, not entirely explained by oncologic therapy, are associated [13]. Further research on the connection of LGLL and solid tumors, as well as on the clinical utility of LGLL screening in cancer patients with unexplained severe/prolonged cytopenias, is needed.

## 5. Solid Organ Transplantation

Solid organ transplantation is a well-known condition associated with secondary lymphoproliferative disease. Allograft-driven immune response, immunosuppressive treatments and concurrent infections create a favorable scenario for clonal lymphoid expansion. Most post-transplantation lymphoproliferative disorders (PTLD) are B-cell malignancies associated with Epstein-Barr virus (EBV) [156], whereas T/NK-cell PTLD accounts for approximately 5% of the total [157].

The exact prevalence of LGLL in solid organ transplant recipients remains unknown [158]. Most of the available evidence derives from case reports [159,160,161,162] and case series [163]. The first article informing about this association was published in 1995 [159] Feher et al. described a T-LGLL in a patient following liver transplantation. To our knowledge, LGLL has been detected in a total of 50 solid organ transplant recipients, in 44 of whom the type of graft has been specified: renal (*N* = 26), cardiac (*N* = 12), hepatic (*N* = 4), and combined transplants (*N* = 2). The median time for LGLL development after transplantation, estimated by Alfano et al., was 10 years [164]. Anemia has been reported as the most usual presentation (50% approximately). PRCA and AIHA have been rarely communicated [160,162,164]. In a work conducted by Awada et al. in 2020, *STAT3* mutations were identified in 2/13 (15%) patients with LGLL after solid organ transplantation [163].

As regards clinical implications, LGLL diagnosis in solid organ transplant recipients encompasses many controversies. A major concern is the prognostic impact of LGLL in this context. Although not completely known, it does not seem that LGLL diagnosis implies poorer clinical outcomes, and the proportion of patients requiring specific therapy is similar to that described in non-transplanted LGLL patients (40–60%) [12]. However, it is important to note that the majority of transplant recipients are usually on immunosuppressive therapy, which is known to be an effective treatment for LGLL [165,166]. Kataria et al. reported a single-center series of four solid organ transplantation patients that developed LGLL with severe transfusion-dependent anemia. Two of the four patients had substantial hematologic response with oral cyclophosphamide therapy [158].

## 6. Post-Allogeneic Hematopoietic Stem Cell Transplantation

The transmission of tumor cells from the donor to the recipient after solid organ transplantation has been described sporadically [167,168]. A national survey of all the organ transplant recipients in the United Kingdom over 10 years found the incidence of donor-transmitted cancer to be less than 0.03% [169]. To date, and as far as we know, no inadvertent transfer of LGL leukemia cells from donors to recipients has been described. Even when the LGLL onset was described early after engraftment (at 3 months), no neoplastic clone was detected in the graft [170]. Thus, in this review, described LGL clonal expansions are donor-derived, not transmitted. Another key point: PB donor T CD8+ LGL expansions after allogeneic hematopoietic stem cell transplantation (alloHSCT), derived from the inoculum in the initial phase of immune recovery or until T-cell compartment reconstitution, are a well-studied physiological phenomenon [171]. It is the persistence of this expansion 1.5–2 years after transplantation, mainly, that poses a challenge to discern its benign or malignant nature.

Persistent expansions of LGL in alloHSCT recipients have been recurrently described, from polyclonal and self-limiting in nature, through clinically indolent monoclonal or oligoclonal lymphocytosis, to even true LGLL [165,172]. Very different percentages of patients affected by this type of expansion after alloHSCT have been reported, ranging from 0.5 to 18.4%. These differences are attributable, in part, to the use of different definitions, longitudinal or cross-sectional assessment, and the incorporation of new molecular and immunophenotypic techniques in the follow-up of post-alloHSCT patients [172]. Likewise, the time interval between the transplant and the onset, as well as the grade of lymphocytosis, are very heterogeneous, ranging from 1 to 61 months, and 0.6 to 11.5 × 10^9^/L, respectively (Table 4).

Though most large series of LGL expansions after alloHSCT found these cases asymptomatic and with improved outcomes, the occasional description of cases with aggressive behavior [181] has generated some uncertainty regarding how to tell the physiological event from the neoplastic one, and whether it impacts the patient´s outcome.

Dolstra et al. first showed how an expansion of CD8+ CD57+ T cells after alloHSCT was related with a low incidence of relapse, suggesting an antileukemic activity of this population [182]. Mohty et al. also pointed toward this antitumor effect in their series, as five out of six high-risk patients achieved a long-term complete remission concomitant or following LGL expansion [172]. This lower risk of relapse was shown to be statistically significant in some following studies [174]. In our work, none of the 14 patients with an absolute clonal CD8+ CD57+ T expansion relapsed, while 22% without these quantitative expansions did (*p* = 0.04) [176].

Two large series have also reported an improvement in OS for those patients with quantitative, persistent, LGL expansions. In 418 allotransplanted patients, Kim et al. showed that cases with LGL lymphocytosis had an OS advantage, lower non-relapse mortality, and lower relapse incidence (in each case, *p* < 0.001) [174]. These findings have also been reproduced by Zhao et al. in a recent series of 359 cases [179].

These expansions are not exclusive to PB or BM alloHSCT. They also appear after unrelated cord blood (UBC) alloHSCT in adults, where persistent B cell lymphocytosis is seen more frequently than in PB or BM alloHSCT. In their work, Les Bris et al. reported absolute lymphocytosis after UCB alloHSCT in 21/85 (24.7%) patients, of whom nine had T-LGL phenotype (three of them were also monoclonal), and one was an NK-LGL expansion. Authors found that absolute lymphocytosis after UCB alloHSCT was associated with favorable outcomes, but B, T and NK cell lymphocytosis were grouped for this analysis, precluding extraction of a specific conclusion on the impact of T and NK cell expansion [178].

Other studies approached post-alloHSCT LGL expansions in a different manner. Recently, Messmer et al. reported a retrospective analysis of 150 unexplained cytopenias after alloHSCT, finding 70 patients with an LGL expansion. Those cases did not show improved outcomes compared to those with cytopenias without the presence of LGL expansions. Forty-three percent of these LGL expansion patients received immunosuppressive therapy, mainly corticosteroids, effectively improving cytopenias [180]. Finally, Awada et al. recently found 15 out of 246 T-LGLL cases that developed after solid organ transplant (*N* = 8) or HSCT (autoHSCT, *N* = 3, alloHSCT, *N* = 4). They found that cytopenias were more frequent among de novo T-LGLL patients, while post-transplantation T-LGLL cases showed higher lymphocyte counts. Interestingly, 2/13 (15%) patients with post-transplantation T-LGLL had acquired a *STAT3* mutation, as noted above, and in both cases after solid organ transplantation [163]. As far as we know, no donor-derived LGL expansion after alloHSCT has been reported to carry a *STAT3* somatic mutation. In our study, even among the 14 cases with absolute persistent LGL expansions, all fulfilling a WHO-defined T-LGLL diagnosis, no mutations in exon 21 of the *STAT3* gene were found. The asymptomatic nature of these cases and the significantly lower relapse rate of the disease that motivated the alloHSCT were also noted [176].

The ultimate cause or triggering factor for the appearance of these donor-derived persistent lymphocyte expansions is not fully elucidated. Variables such as chronic antigenic stimulation due to viral infection, graft versus host disease (GVHD), and/or decreased immune surveillance secondary to immunosuppressive treatments are considered [182]. Some authors propose a two-phase mechanism with an initial polyclonal proliferation that later becomes monoclonal due to secondary events, such as CMV reactivation [183]. Mohty et al. described how, at the exact time of CMV infection, there were no abnormal lymphoid cells in PB smears, with the median time between CMV infection and detection of LGL expansion being 336 days (range 66–399 days). However, at the time of diagnosis of LGL lymphocytosis, no patient had a concurrent CMV infection [172]. In another study, seropositivity for CMV IgG was higher in patients with the presence of an LGL expansion than in those who did not present it [174]. Accordingly, Nann-Rütti et al. showed that 9 of 14 patients with LGL expansion (64%) had previously had CMV reactivation versus 43 of 201 patients (24%) among those without it [173]. Mendes et al. evaluated weekly the CD57+/CD28+/CD8+ T compartment during the first 120 days post-alloHSCT, finding a strong correlation between the expansion of this T population and CMV viremia [184]. In relation to other viruses, the fact that EBV DNA has not been detected in samples from patients diagnosed with LGG leukemia after alloHSCT, together with the fact that T cells do not normally express the CD21 receptor against EBV, suggest that this microorganism does not play a predominant role in the development of these expansions [185]. Sabnani et al. reported negative polymerase chain reaction (PCR) tests against human T-cell lymphotropic virus type 1 (HTLV-1) in patients with T-LGLL after solid organ transplantation [165].

In transplant recipients, viruses are not the only potential cause of persistent antigenic stimulation; rather, it can be induced by the cells of the inoculum. However, this pathogenic mechanism finds a strong counter-argument in the described series of expansion of LGL after autologous transplantation of hematopoietic stem cells where foreign antigens are not present [186,187].

The relationship between LGL expansions and GVHD is debated. Kim et al. showed a statistically significant association between those patients who presented LGL lymphocytosis and the development of chronic GVHD [174]. In another study, this relationship was reached in those patients who had developed acute GVHD [173]. In our 2016 analysis of 154 adult patients with a long-term post-alloHSCT follow-up, persistent relative LGL expansions were frequent (49%) and related with acute GVHD, while persistent absolute LGL expansions were scarce (9%) and related to chronic GVHD [176].

An unsought effect of immunosuppression is the loss of innate defense mechanisms that prevent the development of lymphoproliferative diseases [188]. Mohty et al., in one of their studies, showed that LGL lymphocytosis was more frequent in patients who had received a reduced-intensity conditioning regimen compared to those who received a traditional myeloablative regimen [172]. The authors propose two explanations for this phenomenon. On the one hand, a reduced intensity conditioning could lead to an immune balance that favors the appearance of cytotoxic lymphocytes. On the other hand, this type of regimen allows grafting with less procedure-related toxicity, but this setting is associated with a higher rate of viral infections [189].

Though no *STAT3/STAT5b* mutations have been found in alloHSCT recipients with LGL expansions to date, Liang et al. found that a significant subset of such patients showed nuclear expression of phosphorylated STAT3 protein by immunohistochemistry, which has been associated with constitutive activation of the STAT3 pathway in de novo LGLL, even in the absence of somatic mutations [60,190]. This finding supports the notion that such cases may share a pathogenesis similar to those of non-transplantation related cases.

## 7. Conclusions

LGLL diagnosis is hampered by the fact that reactive persistent clonal LGL expansions may fulfill LGLL criteria. The discovery of acquired mutations in STAT3/STAT5b emerged as a defining criterion for truly neoplastic cases. However, clinical differences between mutated and wild-type cases have not been proved to be categorical, and recently, somatic *STAT3* mutations have been detected in CD8 + T cells of healthy blood donors carrying human T-cell leukemia virus type 2 [3]. Thus, in our opinion, the question remains open. This review outlines clinical settings where the presence of persistent LGL clonal expansions can be detected/suspected. The awareness of these associations and current and future research will contribute to discern whether these persistently expanded clones are cause or consequence of the concomitant clinical settings, and more importantly, when they should be targeted.

## Figures and Tables

**Figure 1 cancers-14-01340-f001:**
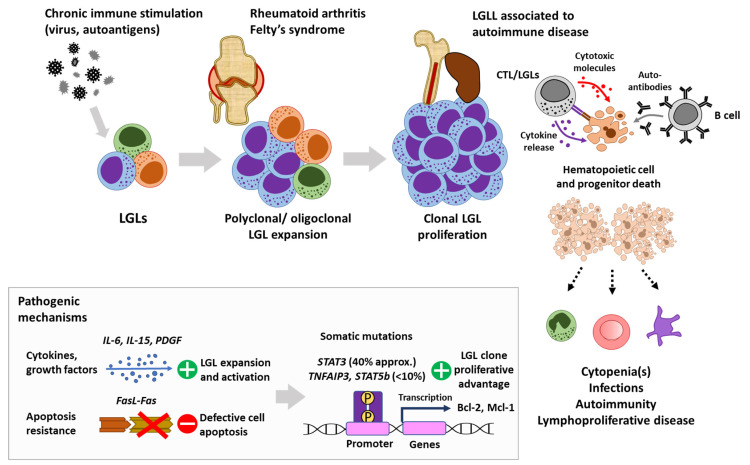
Proposed common etiopathogenesis for LGLL associated with autoimmune disease. FasL: fas ligand. IL: interleukin. PDGF: platelet-derived growth factor. LGL: large granular lymphocytes. LGLL: large granular lymphocytic leukemia. *STAT*: signal transducer and activator of transcription. *TNFAIP3*: tumor necrosis factor alpha-induced protein 3.

**Table 1 cancers-14-01340-t001:** Clinical and laboratory characteristics of three main LGLL cohorts (*N* > 200).

Cohort, Year	Bareu, 2010 [12]	Sanikommu, 2018 [14]	Dong, 2021 [13]
*N* of patients	229	204	319
Median age, y.o.	59	63	65
Sex, *N* females (%)	125 (54.6%)	94 (46.1%)	149 (46.7%)
LGLL characterization			
LGL count, mean (×10^9^/L)	1.71	1.74	0.94
LGL immunophenotype			
T-LGLL, *N* (%)	201 (87.8%)	183 (90.0%)	295 (92.5%)
NK-LGLL, *N* (%)	28 (12.2%)	21 (10.0%)	24 (7.5%)
*STAT3* mutation, *N* (%)	-	66/183 (36.0%)	10/25 (40.0%)
Cytopenia(s)			
Neutropenia, *N* (%)	135 (59.0%)	93 (46.0%)	131 (41.1%)
Neutropenia < 0.5 × 10^9^/L, *N* (%)	56 (24.5%)	36 (17.0%)	4 (16.9%)
Anemia, *N* (%)	55 (26.3%)	79 (40.0%)	132 (41.4%)
AIHA (DAT+), *N* (%)	10 (4.4%)	-	13 (4.1%)
Thrombocytopenia, *N* (%)	40 (17.5%)	59 (30.0%)	83 (26.0%)
ITP, *N* (%)	1 (0.4%)	-	16 (5.0%)
Splenomegaly, *N* (%)	55 (24.0%)	49 (24.0%)	91 (28.5%)
Recurrent infections, *N* (%)	51 (22.3%)	-	-
Serologic autoimmune markers			
Rheumatoid factor, *N* (%)	33/79 (41.8%)	-	41/106 (38.7%)
Antinuclear antibodies, *N* (%)	39/78 (50.0%)	-	25/114 (21.9%)
Associated autoimmune disease			
Autoimmune disease, all *N* (%)	74 (32.3%)	51 (25.0%)	83 (26.0%)
Rheumatoid arthritis, *N* (%)	38 (16.6%)	31 (15.2%)	51 (16.0%)
Associated neoplasms			
Hematological neoplasm, *N* (%)	22 (9.6%)	39 (19.1%)	59 (18.5%)
Solid tumor, *N* (%)	10 (4.4%)	30 (14.7%)	66 (20.7%)
Need for treatment, *N* (%)	100 (44.0%)	118 (58.0%)	181 (56.7%)

AIHA: autoimmune hemolytic anemia. DAT+: positive direct antiglobulin test. ITP: immune thrombocytopenic purpura. LGLL: large granular lymphocytic leukemia. N: number. y.o.: years old.

**Table 2 cancers-14-01340-t002:** List of autoimmune conditions and prevalence of the association with LGLL.

Rheumatic Diseases	Prevalence	Reference
Rheumatoid arthritis	11–36%	[6,11,12,13,14]
Felty’s syndrome	~5%	[11,12,13]
Sjögren’s syndrome (and subclinical forms)	2–27%	[12,42]
Systemic lupus erythematosus	~2%	[43,44,45,46]
Systemic vasculitis	2–3%	[47]
Behçet disease	Rare	[48]
Polymyalgia rheumatica	Rare	[11,13]
Rhizomelic pseudo polyarthritis	Rare	[12]
Inflammatory arthritis, not otherwise specified	Rare	[12]
Inclusion body myositis	Rare	[49]
Organ-specific autoimmune disease		
Endocrinopathies		
Hashimoto’s thyroiditis (and subclinical forms)	2–14%	[11,12,50]
Grave’s disease	Rare	[51]
Cushing disease	Rare	[51]
Polyglandular autoimmune syndrome	Rare	[12,51]
Gastrointestinal tract diseases		
Inflammatory bowel disease	2–4%	[11,12,13]
Autoimmune gastritis (pernicious anemia)	Rare	[11]
Celiac disease	Rare	[12]
Neurologic diseases		
Polyneuritis	2–3%	[12,47]
Multiple sclerosis	Rare	[52]
Organ-specific vascular diseases		
Precapillary pulmonary hypertension	<0.5%	[53]
Glomerulonephritis	Rare	[12,47]
Cutaneous inflammatory diseases		
Leukocytoclastic vasculitis	Rare	[12,54]
Pyoderma gangrenosum	Rare	[55]
Acquired coagulopathy		
Acquired hemophilia A (anti-FVIII)	Rare	[56,57]
Acquired hypofibrinogenemia (anti-FGN)	Rare	[58]
Multiple coagulation factor inhibitors	Rare	[59]

FGN: fibrinogen. FVIII: coagulation factor VIII. LGLL: large granular lymphocytic leukemia.

**Table 3 cancers-14-01340-t003:** LGLL diagnosis in MDS cohorts.

Study	Global Cohort (N)	Frequency of Concomitant Disease	*STAT3* Sequencing Method	Mutation in *STAT3*	Frequency of *STAT3* Mutation (in Concomitant Cases)	Other Altered Genes
Saunthararajah et al. (2001) [79]	100	(9/100) 9%	NA	NA	NA	NA
Jerez et al. (2013) [82]	367	(24/367) 6.5%	ARMS-PCR	D661Y/Y640F	(9/24) 37.5%	NRAS (50%)
Durrani et al. (2020) [83]	240	(13/240) 5.4%	NGS	D661V	(2/13) 15.4%	CHIP (TET2, ASXL1 or DNMT3A, among others)
Komrokji et al. (2020) [84]	1177	(322/1177) 27.4%	NA	NA	NA	CHIP-related genes (TET2, SF3B1, ASXL1, among others)
Ai et al. (2021) [85]	721	(10/721) 1.4%	NGS	NA	(2/10) 20%	ASXL1 (30%) and STAG2 (30%)

**Table 4 cancers-14-01340-t004:** LGLL and LGL expansions in HSCT series.

Reference	(*N*)	HSCT Type	LGL Expansion Assessed (Definition)	LGL Expansion Incidence %, *N*	Time from Transplant to LGL Expansion Detection	TCR Clonality	LGL Count (10^9^/L) Median (Range)	Impact on Outcome (In Multivariate Analysis)	Associations with Post-AlloHSCT Events	Symptoms/ Signs Attributed to LGL Expansion
Mohty. 2002. [172]	201	Allo	T-LGL lymphocytosis (≥2.0 × 10^9^/L)	3% (6)	295 d. (75–450)	33% oligo./ 67% poly.	2.3 (2.0–4.1)	NA	CMV reactivation RIC	4 of 6 autoimmune manifestations
Nann-Rütti. 2012. [173]	215	Allo	T-LGL lymphocytosis (>3 mo.) (≥3.0 × 10^9^/L) & abnormal CD4/CD8 ratio (<1.0 or >1.5)	7% (14)	25 mo. (3–150)	36% clonal/ 7% oligo./ 57% poly.	2.9 (1.3–11.5)	NA	CMV reactivation	5/14 ANA+ 2/14 Polyclonal hypergamma. 3/14 M-comp.
Kim. 2013. [174]	418	Allo	LGL lymphocytosis 2 of 3: (1) ≥3.0 × 10^9^/L; (2) > 30%LGL; (3) Clonal TCR	18.4% (73 T-LGL, 2 NK and 2 mixed NK/T)	312 d. (26–1840)	90% Clonal	1.6 (range 0.6–2.7)	↑ OS ↓ NRM ↓ Relapse	CMV reactivation cGVHD	4/77 pancytopenia 2/77 proteinuria
Gill. 2012. [175]	1675	Allo (*N* = 1246)	LGL Leukemia	0.3% (4)	9 mo.(3–24)	Clonal	3 (1.9–4.7)	NA	NA	Asymptomatic
		Auto (*N* = 429)	LGL Leukemia	0.7% (3)	28 mo. (6–72)	Clonal	2.4 (1.6–2.9)	NA	NA	
Muñoz-Ballester. 2016. [176]	154	Allo	LGL lymphocytosis (>6 mo.) Relative (T CD8+/CD4+ ratio >1.5)	49% (75 T-LGL)	NA	77% clonal/16% oligo./7% poly.	NA	⊜ OR ⊜ Relapse	aGVHD, CMV viremia	1 of 75 (neutropenia)
			LGL lymphocytosis (>6 mo.) Absolute (≥2.0 × 10^9^/L)	9% (14 T-LGL)	NA	All clonal	2.6 (2.2–3.9)	⊜ OR ↓ Relapse	cGVHD	1 AIHA case, 50% MC and/or ANA+
Martell. 2017. [177]	826	Allo	LGL lymphocytosis (>2–3 mo.) (≥3.0 × 10^9^/L)	14.5% (40 T-LGL, 14 mixed T/NK, 2 NK)	306 d. (18–2175)	Clonal	3.7 ± 0.08 Mean ± SD	↑ OS ⊜ Relapse ↓ NRM	CMV viremia aGVHD cGVHD	1 out of 121 (Anemia)
Le Bris. 2017. [178]	85	UCB Allo	LGL lymphocytosis (≥3.0 × 10^9^/L)	8.5% (9 T-LGL/1 NK)	12.6 mo. (1.4–49)	48% clonal/28% oligo./28%poly.	4.8 (4–9.2)	NA	CMV reactivation	NA
Zhao. 2020. [179]	359	Allo	T-LGL lymphocytosis	4.7% (17 T-LGL)	175 d. (25–763)	82% clonal	NA	↑ OS ↓ NRM ↓ Relapse	CMV Reactivation EB viremia aGVHD	Asymptomatic 7 of 17 autoimmune-related parameters
Messmer. 2021- [180]	1930	Allo	LGL lymphocytosis in patients with unexplained cytopenias	3.6% (65 T-LGL, 5 NK-LGL)	6.4 mo. (1.4–81)	22% clonal/53% oligo/4% poly/and 20% indeterminate	1.1 (0.3–6.8)	⊜ OR ⊜ Relapse	CMV viremia	30 treated with IST (83% corticosteroids): effective at improving LGL associated neutropenia

↑ significantly increased; ↓ significantly reduced; ⊜ No significant differences; NA: not available. LGLL: large granular lymphocyte leukemia: HSCT: hematopoietic stem cell transplantation; d: days; mo.: months; oligo: oligoclonal; poly: polyclonal; NRM, non-related mortality, aGVHD: acute graft versus host disease; cGVHD: chronic graft versus host disease; AHAI, autoimmune hemolytic anemia; MC, monoclonal component, ANA, antinuclear antibodies; IST, immunosuppressive therapy.

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
