# Peer review of "Persistent Large Granular Lymphocyte Clonal Expansions: “The Root of Many Evils”—And of Some Goodness"

_cancers, 2022, doi:10.3390/cancers14051340_

Round 1

Reviewer 1 Report

Bravo-Perez and colleagues outline a fascinating and comprehensive review of clinical, biological and therapeutic aspects of LGL-clonal expansion and relative disorders, highlighting how the boundaries between pathological and reactive LGL activation may be difficult to demark.

They go through a huge amount of literature, providing deep insights on clinical correlations with diseases and phenotypes. This review is incredibly comprehensive and it is difficult to find aspects that have not been touched.

  • On a conceptual level, I would add emphasis on the possible pathogenesis of cytopenia in the dedicated section: (from line 80). The authors could eventually clarify more the biological mechanisms associated with bone marrow failure in LGLL: I) T-cell mediated destruction: for instance, the dominant clones can be able to “casually” detect self-antigens on hematopoietic precursors – link with aplastic anemia - (the reason why not all patients show this phenotype?) II) inflammation – increased secretion of cytokine? Possibly IFN-gamma? III) autoantibodies. Regarding this last, it should be also mentioned that the presence of autoantibodies may be a paraphenomenon of altered/cross-reactive immunity, not necessarily cause of a “humoral” mediated hematopoietic stem cell destruction. Please add useful references, and make examples to help the reader to go through these concepts. It would be worthy also to elaborate on the pathophysiological links between autoimmune anemia. thrombocytopenia and T-lymphoproliferation. if lack of evidence in the literature for these connections it would be interesting to hear the authors’ ideas/thoughts.
  • It is not clear whether in table 2 authors indicate the prevalence of the autoimmune disorder or the prevalence of the association with LGL
  • Figure 1: To avoid confusion I suggest removing the associations Felty/LGL/RA and to consider a figure with the pathophysiology of LGL expansion only. Maybe you can add also a connection with B cell activation/autoantibody production.

  • On a more structural/stylistic level, I suggest ameliorating the wording and the linguistic style in some of the sections, that sometimes appear a bit twisted for the reader (i.e., Lines 53-61: please clarify the statements with shorter and non-incidental sentences). Please check also for minor spellings/punctuation (i.e., Line 41 : add a point between "cause" and "The WHO"; Line 52: you missed a proposition before polyarthritis; check and edit the tables – several typos). Avoid also redundancies (i.e. Line 78 you already mention that in some previous lines).

  •  I would also suggest organizing better the paragraphs: I would avoid three paragraphs on RA and Felty/association with LGL or pathogenesis and I would put instead everything under the same subtitle. Idem for paragraphs related to MDS and allotransplant (too many subheadings).

Author Response

Bravo-Perez and colleagues outline a fascinating and comprehensive review of clinical, biological and therapeutic aspects of LGL-clonal expansion and relative disorders, highlighting how the boundaries between pathological and reactive LGL activation may be difficult to demark.

They go through a huge amount of literature, providing deep insights on clinical correlations with diseases and phenotypes. This review is incredibly comprehensive and it is difficult to find aspects that have not been touched.

  • On a conceptual level I would add emphasis on the possible pathogenesis of cytopenia in the dedicated section: (from line 80). The authors could eventually clarify more the biological mechanisms associated with bone marrow failure in LGLL: I) T-cell mediated destruction: for instance the dominant clones can be able to “casually” detect self-antigens on hematopoietic precursors – link with aplastic anemia - (reason why not all patients show this phenotype?) II) inflammation – increased secretion of cytokine? Possibly IFN-gamma? III) autoantibodies. Regarding this last, it should be also mentioned that the presence of autoantibodies may be a paraphenomenon of altered/crossreactive immunity, not necessarily cause of a “humoral” mediated hematopoietic stem cell destruction. Please add useful references, and make examples to help the reader to go through these concepts. It would be worthy also to elaborate about the pathophysiological links between autoimmune anemia. thrombocytopenia and T-lymphoprolypheration. if lack of evidence in literature for these connection it would be interesting to hearauthors’ ideas/thoughts.

We agree with reviewer 1 that more attention into the pathogenesis of cytopenia(s) was needed. We have reformatted the section “Cytopenia(s) and underlying immune-mediated mechanisms” according to his/her wise suggestions: i) we have listed the pathogenic mechanisms proposed for LGLL-associated cytopenia(s); ii) we have speculated about the exact pathogenic role of the LGL clone in the development of cytopenia(s); iii) we have also discussed about the fact that autoantibodies (or other serologic immune markers) may be not always be pathogenic, but also a paraphenomenon in many cases, and iv) we have cited useful references to support these statements.  We thank reviewer 1 for his/her valuable observations on this section.

It is not clear whether in table 2 authors indicate the prevalence of the autoimmune disorder or the prevalence of the association with LGL.

Table 2 indicates the prevalence of each autoimmune disorder in LGLL. This association is now clearly stated.

Figure 1: To avoid confusion I suggest to remove the associations Felthy/LGL/RA and to consider a figure with pathophysiology of LGL expansion only. Maybe you can add also a connection with B cell activation/autoantibody production.

We have removed the associations of Felthy/LGL/RA from Figure 1 (former Figure 1 A). The amended version of Figure 1 only represents the pathophysiology of LGL expansion (former Figure 1B). We have also included in this revised picture the connection with B cell activation/autoantibody production.

On a more structural/stylistic level I suggest to ameliorate the wording and the linguistic stile in some of the sections, sometime a bit twisted for the reader (i.e., Lines 53-61: I suggest to clarify the statements with shorter and non incidental sentences). Please check also for minor spellings/punctuation (i.e., Line 41 : add a point between cause and The WHO; Line 52: you missed a proposition before polyarthritis; check and edit the tables – several typos). Avoid also redundancies (i.e. Line 78 you already mention that in some previous lines).

Done.

To this end I would suggest also to organize better the paragraphs: I would avoid three paragraphs on RA and Felty/association with LGL or pathogenesis and I would put instead everything under the same subtitle. Idem for paragraphs related to MDS and allotransplant (too many subheadings).

Done.

Reviewer 2 Report

Bravo-Perez et al. provide a comprehensive review of LGLL in various settings. Overall, the paper is well written. My only suggestion would be to include some literature and discussion on how MDS can sometimes be misdiagnosed as LGL and vice versa and to perhaps provide some guidance on how to go about differentiating the two diagnosis.

Author Response

Bravo-Perez et al. provide a comprehensive review of LGLL in various settings. Overall, the paper is well written.

My only suggestion would be to include some literature and discussion on how MDS can sometimes be misdiagnosed as LGL and vice versa and to perhaps provide some guidance on how to go about differentiating the two diagnosis. 

We agree with reviewer 2, and we feel this point adds pragmatic depth to our manuscript. We have incorporated it, as follows:

“In a patient with unexplained cytopenias, the potential presence of an MDS and/or LGLL must be assessed. However, the appropriate final diagnosis may be a challenge, particularly in those cases lacking solid MDS features (typical cytogenetics, excess of blasts, ring sideroblasts) or, in the case of LGLL, lymphocytosis. In addition, both diseases exhibit clinical and pathologic overlap with reactive conditions. It has been shown how STAT3 LGLL mutated clones were missed in MDS patients and, for some of them, reviewing the initial workup found that none met the diagnostic criteria for an MDS.[1][2] On the other hand, as stated above, a LGLL diagnosis does not exclude the presence of a concomitant MDS. In this setting, NGS has a clear beneficial role in routine diagnostic practice. The presence of “myeloid” acquired mutations with a higher variant allelic frequency is supportive for the diagnosis of MDS and the inclusion of STAT3 and STAT5b on the NGS panel provides evaluation for cryptic or unsuspected LGL leukemia.[3][2] Besides, a new flow cytometry approach has shown to be more specific for T malignancies than the TCR rearrangement. The pattern of expression of the constant region 1 of the T-cell receptor β chain (TRBC1) has proved to be accurate and simple method for assessment of the neoplastic nature of Tαβ cells in patients with a suspected T-LGLL.[4][5]”

  1. Jerez, A.; Clemente, M.J.; Makishima, H.; Rajala, H.; Gómez-Seguí, I.; Olson, T.; McGraw, K.; Przychodzen, B.; Kulasekararaj, A.; Afable, M.; et al. STAT3 Mutations Indicate the Presence of Subclinical T-Cell Clones in a Subset of Aplastic Anemia and Myelodysplastic Syndrome Patients. Blood 2013, 122, 2453–2459, doi:10.1182/blood-2013-04-494930.
  2. Morgan, E.A.; Lee, M.N.; DeAngelo, D.J.; Steensma, D.P.; Stone, R.M.; Kuo, F.C.; Aster, J.C.; Gibson, C.J.; Lindsley, R.C. Systematic STAT3 Sequencing in Patients with Unexplained Cytopenias Identifies Unsuspected Large Granular Lymphocytic Leukemia. Blood Adv 2017, 1, 1786–1789, doi:10.1182/bloodadvances.2017011197.
  3. Thol, F.; Platzbecker, U. Do Next-Generation Sequencing Results Drive Diagnostic and Therapeutic Decisions in MDS? Blood Adv 2019, 3, 3449–3453, doi:10.1182/bloodadvances.2018022434.
  4. Novikov, N.D.; Griffin, G.K.; Dudley, G.; Drew, M.; Rojas-Rudilla, V.; Lindeman, N.I.; Dorfman, D.M. Utility of a Simple and Robust Flow Cytometry Assay for Rapid Clonality Testing in Mature Peripheral T-Cell Lymphomas. Am J Clin Pathol 2019, 151, 494–503, doi:10.1093/ajcp/aqy173.
  5. Shi, M.; Olteanu, H.; Jevremovic, D.; He, R.; Viswanatha, D.; Corley, H.; Horna, P. T-Cell Clones of Uncertain Significance Are Highly Prevalent and Show Close Resemblance to T-Cell Large Granular Lymphocytic Leukemia. Implications for Laboratory Diagnostics. Mod Pathol 2020, 33, 2046–2057, doi:10.1038/s41379-020-0568-2.